# Current Downhole Corrosion Control Solutions and Trends in the Oil and Gas Industry: A Review

**DOI:** 10.3390/ma16051795

**Published:** 2023-02-22

**Authors:** Vera A. Solovyeva, Khaled H. Almuhammadi, Wael O. Badeghaish

**Affiliations:** 1Aramco Innovations LLC, 119234 Moscow, Russia; 2Saudi Aramco EXPEC ARC, Dhahran 31311, Saudi Arabia

**Keywords:** corrosion control methods, metal protection solutions, cathodic protection, corrosion resistant alloys, corrosion inhibitors, composites, protective coatings

## Abstract

In the oil and gas industry, the presence of aggressive fluids and gases can cause serious corrosion problems. Multiple solutions have been introduced to the industry to minimize corrosion occurrence probability in recent years. They include cathodic protection, utilization of advanced metallic grades, injection of corrosion inhibitors, replacement of the metal parts with composite solutions, and deposition of protective coatings. This paper will review the advances and developments in the design of corrosion protection solutions. The publication highlights crucial challenges in the oil and gas industry to be solved upon the development of corrosion protection methods. According to the stated challenges, existing protective systems are summarized with emphasis on the features that are essential for oil and gas production. Qualification of corrosion protection performance based on international industrial standards will be depicted in detail for each type of corrosion protection system. Forthcoming challenges for the engineering of next-generation materials for corrosion mitigation are discussed to highlight the trends and forecasts of emerging technology development. We will also discuss the advances in nanomaterial and smart material development, enhanced ecological regulations, and applications of complex multifunctional solutions for corrosion mitigation which have become of great importance in recent decades.

## 1. Introduction

In the oil and gas industry, corrosion-related failures have been reported to constitute over 25% of total safety incidents [1,2]. The problem of corrosion in oilfield environments is predominantly associated with the presence of dissolved acidic gases in reservoir brine such as CO_2_ (causing sweet corrosion) and H_2_S (causing sour corrosion); oxygen-related corrosion is encountered in the water injection systems, other associated reservoir constituents, and some as-manufactured defects on metallic components. The consequences of corrosion have large impacts in oil and gas operations. The corroded tubing/casing string can threaten the well integrity and affect the production rate [3]. A statistical breakdown of oilfield corrosion-related failures is shown in Table 1 [1]. According to NACE [4], in 2013, the global cost of corrosion-related economic losses for all industries is estimated to be USD 2.5 trillion which is 3.4% of global gross domestic product (GDP) including corrosion in the oil and gas production industry where the total annual cost of corrosion is estimated to be USD 1372 billion [5]. Application of best corrosion control practices can save up to 35% of corrosion-related losses, which are up to USD 875 billion annually including USD 463 million annually in downhole tubing expenses, and another USD 320 million in capital expenditures related to corrosion [6]. Effective management of corrosion will contribute to an essential reduction of corrosion costs and extension of the lifetime of safe operations.

Although it is not a new problem, corrosion remains the main obstacle for successful operations and commodity integrity in the oil and gas industry.

Corrosion occurs when water is presented at the system and wets the metal surface. This process is composed of three elements that form the electrochemical cell: an anode, a cathode, and an electrolyte (Figure 1) [3]. The anode is the site of metal corrosion, where it dissolves and generates positively charged metal ions [7]. At the same time, electrons transfer from the anode to the cathode. The cathode is an electron-acceptor that is not be consumed. Since the most commonly used metal in the petroleum industry is carbon steel, the metal of corrosion is iron.

Traditionally, most wells were completed with regular low-carbon steel (J-55 [8], L-80 [8,9] and C-95 [9]) that has always been the primary material used in oil and gas operations because of its strength, stiffness, toughness, and tolerance to high temperatures [3], [10]. However, as most of the metals are susceptible to corrosion, it also causes steel degradation that results in the loss of strength and ductility, reduction in the thickness of the pipes, and sometimes even ultimate failure [11]. Such consequences are especially fast and noticeable in a harsh operational environment with a high partial pressure of CO_2_ and H_2_S, at high concentrations of chloride ions, or in the presence of elemental sulfur (S^0^).

Sweet corrosion, which occurs when carbon dioxide is dissolved in water to produce carbonic acid, promotes the electrochemical reaction between the steel surface and the contacting aqueous phase. FeCO_3_ is the main corrosion product formed in a sweet corrosion environment. Iron carbonate forms a semi-protective film on top of the steel surface [12]. At a high partial pressure of CO_2_ and elevated temperatures, the solubility of FeCO_3_ decreases leading to precipitation of the iron carbonate protective scale that prevents corrosion; at low temperatures, the corrosion rate progress [1].

Sour corrosion, on the other hands, takes place as hydrogen sulfide (H_2_S) dissolves in water and reacts with steel to form iron sulfide (FeS) [3]. FeS has limited solubility and forms a semi-protective layer on the steel surface. However, iron sulfide itself works as a cathode to steel, and such precipitate will form an electrochemical potential difference and accelerate the corrosion. In addition to the electrochemical corrosion caused by H_2_S, the most harmful effect of H_2_S is the possibility of causing mechanical chemical corrosion, such as sulfide stress cracking (SSC) [13] and hydrogen-induced cracking (HIC) [3,7,8]. SSC provides the highest risk for OCTG due to the fast and catastrophic failures it causes to the system [5]. SSC is a form of hydrogen embrittlement that occurs in high-strength steels and in localized hard zones in weldment of susceptible metals under certain stress conditions. It depends on the amount of atomic hydrogen presented in the metal lattice as well as on the metal composition, microstructure, and total stress level applied to steel. HIC is a cathodic cracking mechanism that takes place without external stress when steel reacts with hydrogen sulfide and forms iron sulfide and atomic hydrogen. Atomic hydrogen either diffuses into bulk metal or recombines at the metal surface to form H_2_. Dissolved aqueous H_2_S prevents hydrogen recombination at the surface [14]. Therefore, the part of atomic hydrogen that diffuses into the metal matrix is essential; it causes hydrogen embrittlement and cracking. Hydrogen-induced cracking is caused by the blistering of metal due to a high concentration of hydrogen that collects and recombines at inclusions or impurities of steel.

Preventing corrosion in the oil industry is essential to the overall functional integrity of the system. Fast development of smart materials and nanomaterials would allow us to achieve this goal. To bridge the lack of a comprehensive summary of classical and novel methods of corrosion protection and their testing techniques, this work aims to review the most common approaches to mitigate corrosion in downhole applications and summarize the general qualification standards and practice for evaluation of each type of corrosion protection. The survey of corrosion control methods includes application of protective coatings, electrochemical corrosion mitigation via cathodic protection, utilization of corrosion-resistant alloys, and injection of corrosion inhibitors. The overview of each protection method includes discussions of the methods and their quality evaluation. Further, we will outline new trends and aspects to consider in the development of novel anticorrosive materials, highlighting the most recent trends and achievements of material science in the development of corrosion prevention solutions focused on the current industrial challenges. To conclude this work, we will provide our view on future directions for the development of advanced anticorrosive materials with special emphasis on self-healing agents, nanocomposites, and environmentally friendly formulations.

## 2. Corrosion Control Methods

Commonly, corrosion control focuses on materials and environments. Corrosion control methods aim to eliminate the elements of corrosion (an anode, a cathode, and an electrolyte) or preventing these elements from reacting with one another [11]. Corrosion protection could be active or passive. Active protection is the protection that works by taking effective control of the process of corrosion, which includes the modeling of the system, material selection, and general design. Passive protection on the other hand means protection where corrosion is removed or its rate is reduced, by isolation of the material from the corrosive media through the application of barrier layers of coatings, imposing cathodic protection, injection of corrosion inhibitors, and usage of non-metallic composites for downhole equipment protection [15]. Among the active protection methods, selection of appropriate corrosion-resistant alloys and substitution of the metal parts with composites should be noted. The classification of methods utilized for the protection of metals against corrosion is shown in Figure 2.

Below we will focus on each type of corrosion control method in detail along with their qualifications.

### 2.1. Corrosion Control by Protective Coatings

External and internal coating deposition is the most cost-effective and widely used method to prevent corrosion and ensure the integrity of the downhole Oil Country Tubular Goods (OCTG). Corrosion control by coatings and linings (of interior surfaces) is a passive protection. Protective coatings may possess various characteristics, but they have distinct advantages over the other methods of corrosion control. These include:Ease of application;Ease of storage and handling;Range of acceptable ambient conditions;Economics;Ease of restoration.

#### 2.1.1. Basic Principles

Corrosion occurs at the steel interface and this problem is the focus of surface engineering (SE). Surface engineering, which is a multidisciplinary research area, aims to overcome issues of degradation and failure of equipment that are caused by surface processes such as wear, corrosion, and erosion. Basic principles of surface engineering include various types of surface modifications (i.e., surface hardening, plasma thermochemical processing, laser processing) and coating application.

#### 2.1.2. Types of Coatings

The basic function of coatings is to provide a barrier of protection to a metal surface, insulating it from contact with moisture, electrolytes, corrosive gases (oxygen, sour gases), and environmental threats like fluids flow, cuts, and contact with soil. Functional coatings would be selected based on specific functions such as mechanical protection [16], antifouling action [17], corrosion resistance [18], and others. The broad class of anticorrosive coatings is designed to eliminate corrosion. Depending on the surface to protect, the coatings could be external and internal; fusion-bond epoxies (FBE) could protect both internal and external surfaces of tubulars [19,20]. 

The external coating creates a barrier between the tubular and the external environment such as moisture, seawater, fouling, rocks and soils, cement casings, and so on. These coatings often provide improved mechanical protection. 

The internal coating protects the tubular internal wall from the flow media, increases pipe smoothness and flow characteristics, prevents the corrosive action of acids, moisture, electrolytes, hydrocarbons, and sour gas exposure, as well as reduces scale and asphalting deposition. 

For downhole applications, several types of coatings are used that can be divided into metallic, non-metallic, and mixed type coatings. Non-metallic coatings can be either organic or inorganic (Figure 3).

##### Metallic Coatings

Metallic coatings are thin layers of metals deposited onto the steel surface. Insulative coatings consist of less reactive metals which are used to coat the base metal, for example, nickel-plated steel. A sacrificial coating consists of a more reactive metal that is deposited on the steel surface, e.g., zinc-coated steel. Important metals used for coating deposition are nickel, zinc, chromium, tin, aluminum, copper, and others [15]. In addition to metals, alloys could also be deposited as metal coatings; thus, NiAl, FeCr, FeCoCr, FeCoCrNi, and FeCrNi coatings [21] have been suggested for steel protection. The advantages of metallic coatings are remarkable corrosion resistance, excellent mechanical properties, uniform film deposition, very good wear resistance, high hardness, abrasion resistance, high adhesion with excellent range of temperature and pressure resistance, as well as tunable acid stability. Conventionally, most of the metallic coatings would be applied via methods such as electroplating, cladding, electroless coatings, spraying, hot dipping, galvanizing, anodizing, plating, sputtering, and chemical vapor deposition. A few of the most common methods of metal coating deposition used in the oil and gas industry are:Hot-dip galvanizing—immersion of steel substrate into a bath of molten zinc [22]. Additionally, hot dipping technology could be applied for aluminum, zinc–aluminum alloys, tin, antimony, lead, and other low-melting-point metals and alloys.Electroplating—deposition of a metal coating on top of other metals via reduction of the cations of the coating metal upon application of a direct external electric current. The most widely used electroplated metals are zinc, nickel, copper, and chromium [22].Electroless nickel plating [23]—autocatalytic chemical reduction that deposits a uniformly thick layer of amorphous nickel or nickel–phosphorus alloy on top of the substrate. The process occurs upon dipping the substrate into a solution of nickel salt and the reducing phosphorus agent (commonly sodium hypophosphite) with no use of an external current. The content of the phosphorus governs the coating’s microstructure, ductility, and rate of deposition. Rod pumps, packers, mud pumps, collars, couplings, safety valves, and other equipment are coated with electroless nickel plating in petrochemical applications due to its great chemical resistance [23].Thermal spraying—atomization of molten or semi-molten droplets by the impingement of quickly moving and continuously flowing atomization gas onto the prepared base surface [24].

Numerous advantages of the metallic coatings listed above led to the extensive development of advanced coating deposition methods, inventions of numerous coating compositions, and coatings specialization; however, their cost is about 5–10 times more expensive than carbon steel, and their application techniques involve toxic chemicals. Considering rising environmental regulations on heavy metals, cyanides, and toxic gas pollution, the engineering of an alternative type of coating is essential.

##### Non-Metallic Coatings

Thermoplastic and thermosets are the most common key components of the non-metallic coatings that have been used for rehabilitation of the existing OCTG and for deposition on new products. Non-metallic coatings can be subdivided into organic (polymers) and inorganic (oxides, phosphates, nitrides, sulfides, carbides, etc.) [25].

##### Organic Coatings

Organic coatings consist of mixtures of binder, pigments, fillers, solvents, and additives where the binder is the major component of the coating. It determines the nature of the coating system that can be epoxy, phenolic, polyolefins, polyurethanes, polyamides, combinations thereof, and others. Fillers serve specific coating functions, for example, extended anticorrosive action such as zinc-rich epoxy [26] where zinc is used as a sacrificial agent or as corrosion-inhibiting zinc phosphate fillers [27]. Solvents are optional in the coating composition but for the multi-component liquid formulations, the solvents are the components that help in uniformly dispersing all the other ingredients in the formulation. Additives can serve various purposes; they may be present as reinforcement agents such as carbon black, graphene [28], mica, clay, aramid, and carbon and glass fibers [29] to provide strength and additional mechanical stability. Additionally, they may act as curing or flow control agents or many other functions. 

There is a wide variety of organic OCTG internal coatings including epoxy, FBE, phenolic, modified epoxy–phenolic, modified urethanes, nylon, [30] and composite fiberglass coatings.

**Epoxy** coatings consist of diglycidyl ethers of bisphenol A [31], F, and other bisphenol moieties. The curing agents, such as aromatic (methylene dianiline) and aliphatic diamines (isophorone diamine) and polyamines, drastically influence the nature and the properties of the final coating. Basic epoxies are thermosets that are stable until 175 °F and are constructed from bisphenol A and epichlorohydrin building blocks. The common examples of classical epoxies are Epon, Durcon, and Araldite [30]. These coatings possess good chemical resistance and excellent flexibility in thick films of 254–381 μm (10–15 mils) [32].

**Phenolic** coatings are thermosets based on phenol–formaldehyde with the most well-known example being Bakelite [30]. These coatings are stabile up to 400 °F and are chemically resistant to brine containing CO_2_ and oxygen, or hydrochloric and hydrofluoric acids. These coatings possess high rigidity and mechanical properties but are not as flexible as epoxies. 

**Novolac** resins are highly crosslinked phenol formaldehyde resins with a ratio of phenol to formaldehyde of less than one. It often contains cresol moieties instead of or in addition to phenol and is produced by acid-catalyzed polymerization. Hexamethylenetetramine is a common hardener agent to crosslink novolacs.

**Epoxy–phenolic** resins are phenolic resins with the aromatic hydroxyl position modified with epichlorohydrin to introduce an epoxide functional group. This resin possesses a high degree of crosslinking compared to the phenolic analog due to additional epoxy functions.

**Modified epoxy–phenolic** coatings consist of epoxy–phenolic resin components modified with inorganic fillers to increase abrasion resistance [33]. These coatings possess general resistance to chemicals, are stable up to 400 °F/204 °C, and can be applied in thin films that are 127–228.6 μm (5–9 mils) thick [32].

**Urethanes** or polyurethanes could be both thermoset and thermoplastic coatings. They are formed upon reaction of di- or triisocyanates with polyols to form alternating copolymers linked by carbamate units. For high performance coatings, fluorinated polyols are often used. 

**Modified urethanes** are polyurethane matrices modified with inorganic fillers like ceramics and/or glass particles to increase wear, and corrosion and abrasion resistance. Urethane resin coatings have high resistance to water, oil, and hydrocarbons [34] as well as good elastic properties and are stable up to 400 °F/204 °C if applied as 127–228.6 μm (5–9 mils)-thick films [32]. 

**Nylon** is a family of thermoplastic polyamide coatings. They possess chemical resistance to water, carbon dioxide, and hydrocarbons with temperature stability up to 107 °C/225 °F [30].

**Fluoropolymer**-based (FP) thermoplastic downhole coatings protect OCTG at HTHP conditions of 8000 psi, 350 °F in sweet and sour media [35]. New generations of engineered FP superhydrophobic coatings additionally prevent the formation of organic (asphaltenes and paraffins) and inorganic (carbonates, BaSO_4_) deposits on the surface of production pipes for both oil and gas wells. Nowadays, FP-based coating systems could address the challenges that previously could be resolved only with the utilization of CRAs. Current FP coatings are recommended for wells with a harsh corrosive environment, pressures above 10,000 psi, temperatures above 392 °F/200 °C, content of H_2_S above 15%, and high salinity of production brine.

Some of the abovementioned OCTG coatings, their stability and applications in oil production are listed in Table 2.

The great progress in polymer design and synthesis achieved in the recent decades led to a breakthrough in the properties of organic coating formulations that brought their performance close to the reliability and robustness of metallic materials. Moreover, these novel organic coatings still retain their major advantages in cost efficiency, ease of deposition and repair, as well as being light weight and having tunable elasticity. Considering the number and varieties of commercially available polymers and the vast possibilities for their modifications with advanced fillers, composite coating formulations could reach further to overcome the metals. Furthermore, tuning of organic matrices with nanomaterials allows for noticeable improvements in the performance even at a low filler loading.

##### Inorganic Coatings

The main methods of inorganic coating include oxidation, phosphating, enameling [15], and others. Depending on the metal nature and the spare part specifics, the optimal coating and method of deposition should be chosen. Oxidation involves coating of a metal with a surface oxide film by heating the metal at high temperatures in the presence of oxygen, by chemical oxidation treatments with acids, or by anodic oxidation in an electrolytic cell. Metals which can form non-porous oxide layers to convert steel surfaces to be more passive are zinc, aluminum, and magnesium [39]. For austenitic and martensitic stainless steels, corrosion resistance depends on the content of chromium and the purity and uniformity of surface chromium oxide layers [39,40]. 

Phosphating involves coating of steel with a layer of porous crystalline mixed phosphates [41]. The nature of the coating could include hydrated zinc phosphate (Zn_3_(PO_4_)_2_ * 4H_2_O, hopeite), iron and mixed iron–zinc phosphates (Zn_2_Fe(PO_4_)_2_ * 4H_2_O, phosphophyllite), mixed iron–manganese phosphates, and/or zinc and other phosphates [42]. Phosphates are porous, thus offering only partial corrosion protection, and therefore are being used as a base for liquid coatings that prevent the spread of rust under the oils and paints. There are a number of the patented formulations of phosphate conversion coatings [43,44] that could be applied via brushing, spraying, or immersion of the steels into the orthophosphate solution with salts of zinc, magnesium, or manganese [45] in the presence of phosphoric acid. In oil production, phosphate conversion coatings are used to improve the sealing ability of casing connections [46].

Enameling implies coating of the substrate with a glass layer by dipping the metal into a suspension of powdered glass followed by high temperature melting of the glass on the metal surface. Groundless enamels are applicable for the internal surfaces of the pipes and require forcing the slip under pressure into the internal space of the pipes, followed by discharge. Thus, a uniform layer of enamel forms and slips over the entire length of the pipe [47]. Porcelain enamel, as an inorganic material, is chemically bonded to substrate metals by fusing glass frits at a temperature of 750~850 °C. It provides good chemical stability and corrosion resistance, as well as excellent resistance to abrasion and thermal shocks in extreme erosion environments, far outperforming epoxy coatings [48].

High performance zinc-rich silicate inorganic coatings are well known in the oil production industry since 1970s and have been successfully used for the protection of steel against corrosion in extreme environments [49]. These systems offer good if not the best thermal stability among the liquid coatings that are stable up to 400 °C. For deposition of zinc-rich silicates, conventional spaying is used. The conductive nature of the silicate binder in combination with zinc, provides perfect cathodic protection, metal-like hardness, and excellent adhesion to steel; however, these coatings cannot resist pHs other than the range of 5–10. These coatings find their extended use in protective and reinforcement coatings for drill pipes [50].

The other types of inorganic coatings, such as boronized ceramic-based tribo-corrosion resistant systems, were proposed and showed a high efficiency of protection of oil production equipment such as artificial lift systems, centralizers, valves, and some other complex-shaped components of engineering tools [51].

##### Mixed Inorganic–Organic Coatings

The addition of inorganic species into organic coatings improve their thermal, mechanical, and wear resistance properties in addition to providing galvanic protection as was shown for zinc-rich epoxy coatings [52]. Moreover, inorganic mesoporous materials such as TiO_2_ can be surface modified with scale and corrosion inhibitors to serve as an inhibitor carrier and chelating agent within the epoxy matrix to improve the corrosion resistance and anti-scaling properties of the coating [53]. Mesoporous TiO_2_ whiskers were surface modified with ethylenediamine tetra(methylene phosphonic acid) (EDTMPA), a well-known chelating, scale reducing, and anticorrosive agent, and were further loaded with imidazoline corrosion inhibitor into internal pores to yield multifunctional iFTiO_2_(w) carriers [53]. These iFTiO_2_(w) carriers were evenly dispersed within organosilicon epoxy coating and used to coat steel coupons. Outstanding barrier, and scale and corrosion prevention properties were demonstrated for this blend due to synergistic protective effect of EDTMPA and imidazoline-functionalized composite. iFTiO_2_(w)-containing coatings worked up to 35–40 times better compare to non-modified-TiO_2_-loaded epoxy coatings.

The overview of the abovementioned types of protective coatings taking into consideration the economic aspects of their deposition and the ecological trends in green industrial operations allows us to conclude that the most promising and cost-efficient material engineering efforts should be targeted at the design of polymeric and nanocomposite formulations involving deployment of smart and stimuli-responsive additives. Metallic coatings were previously unique due to their superior protection and mechanical resistance but they are becoming less economically feasible compared to emerging engineered polymers and composites. There are broad possibilities for the customization of epoxy, phenolic, and other polymeric matrices towards resistance to specific temperatures, aggressive media, and mechanical impacts, allowing polymeric and nanocomposite formulations to reach the efficiency of metallic coatings with lower costs of deposition. The current tendency in mixed coating engineering is to overcome the gap in the mechanical rigidity and robustness of polymer-based composites and metals. Mixed coatings formulations that combine the advantages of organic and inorganic/ceramic materials with functional nanomaterial additives enable further improvement of their protective properties. A comparison and evaluation of the efficiency of these coatings could be performed based on the qualification techniques that are reviewed below.

#### 2.1.3. Performance Qualification of Coatings

Building on the importance of the coating performance, this section is dedicated to the quality control and coating evaluations. Various methods of nondestructive QC tests including visual and ultrasonic coating thickness evaluations and continuity examinations are performed at manufacturing sites upon OCTG production; however, regular inspection of new products in the market and monitoring of the performance of applied protective coating systems is necessary for maintaining the durability and integrity of the well infrastructure. Recommendations for QC industrial techniques are summarized by major international standards such as ISO, ASTM, NACE, and others. Below we will focus on the most common methods of coating evaluation.

##### Electrochemical Impedance Spectroscopy (EIS)

Electrochemical impedance spectroscopy allows characterization of heterogeneous systems composed of multiple arrays of layers with different electrical and structural properties by applying equivalent circuits. This method is based on the determination of electrical impedance of a sample as a function of the frequency of applied alternating current [54]. This technique is a non-destructive, online method to control general corrosion, and its inhibition and coating performance in the laboratory. It has no widespread field monitoring application due to localized zone of measurements, the complexity of the results’ analysis, and being difficult to use during wide temperature and operational changes [55]. EIS is a useful lab tool to detect areas of localized active corrosion at the metal–coating interface [56] in order to test the self-healing processes of the coatings, to evaluate the impact of additives like various ions into the coating matrix, and to evaluate the barrier properties of the coatings [57]. These measurements can be used to determine the effects of the water uptake and subsequent coatings plasticizing [58] and others. In a typical experiment, the coated metal surface with an artificial cut in the protective layer is exposed to sea water and immersed with a cathodic current. The changes of impedance are measured relative to a reference electrode (saturated calomel electrode (SCE)) in the electrochemical cell [58]. Ageing and degradation of the coating system is indicated by the decay of the impedance modulus with immersion time. Equivalent circuits are used to model and experimentally verify the EIS data. Several software solutions such as ANALEIS with the modules BASICS and COATFIT [59], PowerSuite [60], and others were developed to simulate and interpret EIS spectra.

##### Adhesion Tests

The strength of a coating’s adherence to the substrate is its principle functional characteristic. Adhesion is a measure of the force per unit area required to remove the coating from the surface [55]. Adhesion strength tests determine what force is required for separation of two adhered parts along the interface. Typically, adhesion tests are based on different methods of separation of a coating from its substrate at a particular load. Below we will list the most essential adhesion tests recommended by ASTM and ISO standards.

**The pull-off test** is utilized for the evaluation of adhesion of single and/or multiple coating systems on a metal surface by application of a tensile stress from a dolly to the coating through the usage of a portable adhesion tester/automatic pull-off adhesion gauge. The stress gradually rises until the dolly with the adhesive layer is removed. Pull-off strength determination includes two protocols: (1) testing if the coated surface remains intact at a defined loading and (2) testing for fracturing that determine the maximal perpendicular force that the surface can bear until the material detaches. The test can be performed in the field and would provide quantitative evaluation data according to ASTM D4541-17 [61], ASTM C633-13(2017) [62], and ISO 4624:2016 [63] techniques. Figure 4 shows the pull-off test description and test instrument kit.

**An X-cut adhesion test** evaluates the adhesion of single or multilayered coating systems to a metal or another coating surface using one stroke from a knife according to ASTM D6677-18 [66]. This technique is applicable both in the laboratory and in the field. The test should be performed with the cut made all the way to the examined substrate surface.

**The scratch test** is standardized by ASTM D7027-20 [67] and the equivalent ISO 19252:2008 [68]. The scratch test measures the adhesion strength of the coating upon applying a scratch machine with an increasing amount of pressure until the adhesive is detached. The commercial scratch test apparatus allows control of the scratch velocity and length. Detection of the critical load upon coating debonding is a quantitative characteristic of the method. Adhesion and cohesion between the coating layers as well as the types of debonding is evaluated.

**The bend test** according to ASTM D522 [69] is based on producing shear stress along the coating interface by bending the tested sample over a round pin as shown in Figure 5.

**The shear test** according to ATSM F1044-05(2017)e1 [70] is used to determine the adhesion and internal cohesion of the metal and phosphate coatings in shear parallel to the surface as shown in Figure 6. Coated specimens are glued against the matching counterpart and load up to failure. This test determines a material’s ability to resist forces that cause the coating system to slide against itself, that is important to provide correct handling of the coated products. Chen [71] provides the methodology of the mechanical evaluation of coating adhesion.

##### Abrasion Resistance

ASTM D4060-19 [73] is a standardized technique to measure the abrasion resistance of organic coatings on a plane rigid surface by the Taber Abraser. The instrument is a single rotary platform with various abrasive wheels that perform accelerating wear examination mimicking fine, coarse, and large particle abrasion. The technique allows determining the wear index, weight loss, and/or wearing cycles per Mil. A lower wear index indicates better abrasion resistance of the coating.

##### Impact Resistance

ASTM D2794-93(2019) [74] describes the technique to qualify the coatings for impact resistance. A standard weight is dropped from a distance onto the coated metal plate to strike an indent that deforms the coating and the substrate. By gradually increasing the distance of the weight drops, the point at which coating failure occurs is determined. Visible inspection of the crack with a magnifier or usage of a pinhole detector or chipping with tape allow identification of the damage.

##### Immersion Test

The standard ASTM D870 [75] covers the evaluation of the stability of the coating system in water at ambient or elevated (up to 95 °C) temperatures upon immersion of the coated specimen. The typical results of the test are “pass” or “fail”, although a degree of failure could also be determined. Performance of the coatings immersed in a 5% sodium hydroxide or 5% sodium chloride solution should be examined at ambient and elevated temperatures according to NACE TM 0174 [76].

##### Cathodic Disbonding Test

Although original ASTM G8-96(2019) [77] was developed to test polymeric pipeline coatings, the method has been successfully applied to evaluate the adhesion of downhole thermoplastic and thermosets coatings [78]. A truncated version of the ASTM G8-96(2019) technique was shown to be effective for the evaluation of the adhesive properties of coatings with various compositions such as polyamide and different types of epoxies. The test exposes the coated metal specimen with a coating perforation to high cathodic polarization followed by physical examination of polymer disbonding around the perforated site and/or the monitoring of the current drawn by the test specimen. The current that is applied as the cathodic protection induces alkalization of the interface between the metal and coating and may lead to cathodic disbonding of the coating [58]. Along with ASTM G8-96(2019) that covers the standard testing of coatings for cathodic disbonding resistance at room temperatures, ASTM G-42-11(2019)e1 [79] describes the standard testing method to determine cathodic disbonding of pipeline coatings at elevated temperatures, and ASTM G95-07(2013) [80] is valid for the testing of coatings where the test cell is cemented to the coated specimen. Physical examination is used for the inspection of the coating contact to the metal surface. The cathodic disbonding test apparatus includes an electrolytic cell, potentiostat, and voltage meter.

##### Torque Testing of External Coatings

The integrity of the external coating is essential to withstanding the downhole environment temperature, water salinity, and chemical compatibility. However, the external coating integrity is especially important when OCTG are moved onto the rig using power tongs. The torque that is applied to the external coating will damage the coating and it will require the use of the fast cure epoxy on the rig to fix the dents or damage to the external coatings. 

External coating qualification using a rotational torque machine is required to evaluate the adhesion upon exposure to an external mechanical impact. The rotational torque machine mimics the actual rig torque and helps to evaluate the external coating integrity on the shop as part of the qualification. This torque will check the resistance of the coating and evaluate the damage/dents that may occur (Figure 7).

As reviewed above, coating qualification and standardization procedures are the special part of coating QC practice that involves a several stages of evaluation. It includes primary laboratory tests as well as the qualifications at the workshop to select superior materials for preliminary field investigations. Stationary field performance of the material during tests provides an estimate of the actual work performance of the system in the dynamic ever-changing conditions in the field. Thus, a wide variety of tests is used for coating evaluations to evaluate the behavior of the coating system during most of the possible challenges that could occur in the field.

### 2.2. Corrosion Control by Cathodic Protection

#### 2.2.1. Basic Principles

Cathodic protection (CP) is an electrochemical corrosion mitigation technology. Its main principal is based on diminishing the corrosion of a metal by making that surface the cathode of an electrochemical cell [82] with the purpose of significantly reducing the corrosion rate of the metal [83]. CP is intended for metal structures permanently submersed into sea waters [84], structures exposed to soil [85], or concrete such as well casings [86].

#### 2.2.2. Cathodic Protection Design

There are two types of cathodic protection that can be applied—galvanic and impressed current cathodic protections. The first one works as electrochemical protection in itself, while the second one requires an external source for the current. 

##### Galvanic Cathodic Protection

A galvanic cathodic protection system consists of a sacrificial anode fixed directly to the metal surface. The sacrificial anode should be made of a highly active metal to prevent the less active metal from corrosion [87]. Galvanic anodes typically could be made of alloys of magnesium, zinc, or aluminum [88]. For some of these metals there is a risk of passivation of galvanic anodes due to the formation of a surface oxide layer that would diminish anode capacity. The galvanic cathodic protection has a limited lifetime until the sacrificial anode is consumed. Once the sacrificial anode is no longer capable of protection, the original metal surface would start to corrode.

##### Impressed Current

The impressed current cathodic protection utilizes a current from an external power source. Such protection converts all anode areas to cathodes, thus eliminating the possibility of an anodic loss of metal (Figure 8) [87].

Since the power is delivered to the electrode from an external power source and is not generated by electrode degradation, this system provides a much longer protection lifespan compared to the sacrificial anode and it enables calibration of current parameters. Impressed current cathodic protection works differently for buried pipes and for the well casing. Buried pipes and pipelines are typically protected with direct current (DC). 

Criteria for the sufficient cathodic protection method are different depending on the type of metal and environment conditions. Formation of acidity and soil resistivity, presence of bacteria, and thermal deviations would require the adjustment of the CP parameters. Below, as an example, we list basic empirical criteria to indicate the corrosion control effectiveness for steel and gray or ductile cast-iron piping systems according to NACE SP0169 [84,85]: A minimum of 100 mV of cathodic polarization between the structure surface and a stabile reference electrode containing electrolyte. Either the formation or the decay of polarization must be measured to satisfy this criterion.A structure-to-electrolyte potential of −850 mV or more negative as measured with respect to a saturated copper/copper sulfate (CSE) reference electrode. This potential may be either a direct measurement of the polarized potential or a current-applied potential.A minimum negative (cathodic) voltage shift of 100 mV or more as determined by interrupting the current and measuring the voltage decay in the earth and metallic parts.

For the cathodic protection of closely spaced well casings due to interference of currents [39], pulse current technology was demonstrated to provide superior results over the conventional DC in certain soils and casing lengths; however, there is no absolute solution and only 20% of the top of the casing part could be protected by CP [85,89,90]. Pulse cathodic protection uses pulse rectifiers to generate short electrical pulses several thousand times per second with a typical frequency range between 1000 and 5000 hertz and an output voltage of 0–300 V [85]. Advantages of this type of protection include: (1) the current requirements are typically significantly lower than in conventional DC; (2) pulse cathodic protection can provide a current to significantly deeper levels than conventional DC; and (3) this system provides more uniform current distribution to the casing [90].

To conclude, CP optimization is based on synergy of electrical engineering and material science. From the point of view of material design, the novelty in the development of cathodic protection systems would rely on the improved composition of sacrificial anodes to avoid their passivation due to formation of oxides, hydroxides, and carbonates on the anode surface. Thus, the recently described galvanic coupling of mild steel with high phosphorous pig iron (HPPI) [91] exhibited efficient cathodic corrosion protection of the steel in concrete, soil, and seawater due to the formation of well-soluble FePO_4_ * 2H_2_O upon anode consumption. Incorporation of more active elements as Al, Mg, and Zn into the HPPI resulted in hybrid anodes that improved the performance of CP even better than ordinary HPPI via cooperative action of electrochemical effects of additives and anode dissolution [92]. Furthermore, composite anodes with the inclusion of metal oxides as Al_2_O_3_, ZnO, MnO_2_, and CeO_2_ to Al–Zn alloys improved uniformity of anode corrosion and thus, increased its galvanic efficiency [93]. Comparison of materials for the cathodic protection efficiency are based on the qualification methods that are reviewed below.

#### 2.2.3. Qualification of Cathodic Protection

##### Qualification of the Sacrificial Anode

For cathodic protection with a sacrificial anode, spectrometric analysis of chemical composition, verification of anode weight, straightness, and dimensions, check of structure and surface irregularities, and testing of electrochemical efficiency and closed-circuit potential should be performed according to DNV and NACE/AMPP standards [82,84,94].

Tests for the screening of the appropriate material for the sacrificial anode may typically use a single parameter (e.g., operating potential at a defined constant current density) as a pass/fail criterion and are normally of a short duration (usually hours) with test specimen weights of few tens to hundreds of grams [95].

Performance tests of sacrificial anodes are of much longer duration (from months to years) and require close mirroring of the anticipated field operational conditions for testing of specimen operating potential for the anode (tens to hundreds of kilograms). 

For chemical quality control tests the anode material is subjected to spark emission spectroscopy of a melted, cut, or acid-dissolved part of the specimen to determine its actual chemical composition.

Exposure testing was first developed to act as a quality check on the efficacy of solution heat treatment of Al–Zn–Sn alloy anodes when a simple capacity test based on measurement of hydrogen evolved under impressed current was used as “go” or “no go” acceptance criteria [95].

##### Qualification of the Impressed Current

For the impressed current CP, the NACE/AMPP standard [96] summarizes the methods of quantitative evaluations of cathodic protection efficiency. It includes methods for voltage drop examination when structure-to-electrolyte potential measurements are made and provides guidance to avoid vulnerable data. The qualification should include examination of the impressed current rectifier, impressed current positive and negative circuit measurements, impressed current structure to soil potential survey, and testing of the continuity of the impressed current [97]. The impressed current continuity survey includes techniques of “fixed cell—moving ground test” and point-to-point continuity test. 

Impressed current positive and negative circuit measurements could be performed with an ammeter or digital multimeter at the systems where individual lead wires for each anode are installed. 

The impressed current structure-to-soil potential survey [88,98] requires a reference electrode and voltmeter. Pipe-to-soil potential measurements are performed with an applied protective current and are recorded over equally distributed test points. Potential vs. test points distance provides data on the efficiency of the CP (Figure 9) [97]. The voltage at the applied and temporarily interrupted impressed current is recorded against the reference electrode as “On” and “Off” potential to determine pipe polarized potentials. The acceptable values of “Instant Off” potential are tabulated [97].

The impressed current continuity survey is a point-to-point test performed with a turned off rectifier; the leads of the voltmeter (at ≥2 volts DC setting) connects to two test structures to evaluate their insulation or continuity via the voltage difference. The continuity is determined if the voltage difference is 1 mV or less. For the entire structure protected with impressed current CP, all points should be continuous [97].

For well casings, the methods for determining the specific current density for optimal CP can be developed according to [99] based on information from:(1)Casing potential profile—The measurements of voltage drop across a portion of well casing in service;(2)Current density calculations based on well completion practices;(3)Mathematical modeling including current attenuation modeling;(4)E-log-I test results.

### 2.3. Corrosion Control by Corrosion Resistant Alloys (CRA)

#### 2.3.1. Basic Principles

Corrosion-resistant alloy is a commonly used term to cover a broad spectrum of alloys with compositions of metals such as Fe, Ni, Cr, Mo, W, Ti, and others, which are capable of efficiently resisting corrosion compared with carbon steel. These alloys are employed in extreme oil and gas production environments that operate at HPHT (pressure exceeding 1000 bar and temperatures over 177 °C) and contain chlorides, CO_2_, and H_2_S [100]. CRAs exhibit high resistance to uniform corrosion due to their passivity. In some cases, if the downhole environment is extreme, the best way to control corrosion is to use CRAs that are more resistant to degradation than structural steel.

#### 2.3.2. Types of Corrosion Resistant Alloys

The most common CRAs materials used in downhole applications [101] are 9CR, 13CR, 316L stainless steel, and Inconel^®^625. 9Cr is a general-purpose chromium–manganese–molybdenum–iron–carbon alloy with an average Cr content of 9%. It is often used for tubular resistant to high pressure–high temperature in sweet environment service. 13Cr (martensitic stainless steel) is a chromium–manganese stainless steel with an average Cr content of 13% that possesses comparatively high resistance to mild sweet and sour environments including exposure to wet CO_2_ containing chlorides, but its exposure to oxygen in the presence of chlorides would result in the pitting of 13Cr as well as its exposure to high concentrations of hydrogen sulfide. 13Cr is used for casing and tubing production and as conventional completion material for sand screens [102]. 316L (austenitic stainless steel) is a chromium–nickel–molybdenum–iron alloy with 0.03% carbon; it possesses great resistance to acidic conditions containing CO_2_ and H_2_S but is susceptible to local corrosion in chloride ion-containing media [103]. 316L has a low carbon content that prevents corrosion caused by welding; it is used for the manufacturing of tubing products [104] as well as for expandable sand control screens [102]. Inconel^®^ 625 is a corrosion and oxidation resistant, nickel–chromium–molybdenum–niobium superalloy used for ESPs, valves, and cladding manufacturing [105]. It has high strength and toughness in temperatures up to 1000 °C and outstanding chemical resistance to chlorides, seawater, alkaline media, oxidizing chemicals, highly acidic CO_2_ and H_2_S environments, and nitric and hydrochloric acids. The chemical composition of the most common downhole tubing and casing materials grades is shown in Table 3 [10].

Since the 1970s, the application of CRAs in the oil and gas industry and data on the selection of appropriate alloys for extreme conditions have been constantly expanding. However, the initial investments cost for CRA utilization has exceed the cost of ordinary carbon steel protected with organic coatings by several times [106]. Thus, careful decisions should be made at the initial stage of well design with balanced considerations of CAPEX and OPEX of implementation of alloys vs. coated carbon steel in combination with corrosion inhibitors. Corrosion-resistant alloys that are used in extreme operational conditions despite their superior performance should be additionally protected with corrosion inhibitors upon production operations including acid treatments that can significantly affect CRA resistance. Sridhar gives useful guidance for the choice of the corrosion resistant alloy according to designated downhole conditions [100]. Until now, the service performance of CRAs and superalloys remains the best for the extreme downhole environment; however, usage of the composites and coated OCTG in broader cases could be more economically viable. 

#### 2.3.3. Qualification of Performance of Corrosion Resistant Alloys

Before use in oil production, CRAs should be subjected to examinations to evaluate their stability in designated service conditions. ANSI/NACE MR0175/ISO 15156 [107] provides a guideline for the selection of CRAs for sour service. ISO 13680 [108] covers a full list of tests that are necessary to qualify tubular products for oil and gas production, including examination of the chemical composition, mechanical characteristics of the CRA products, tensile and hardness tests, impact of flattening, impact test at low temperatures, pitting corrosion test, microstructural examinations detecting ferrite/austenite ratio, dimensional test, drift test, determination of length, straightness, and mass, visual inspection, and non-destructive evaluations. Petersen [109] lists the main types of tests recommended to qualify various classes of alloys.

For sour conditions, the important qualification parameters [110] are environmentally assisted cracking tests that cover stress corrosion cracking (SCC) evaluation, hydrogen embrittlement examination, and localized corrosion tests.

##### Stress Corrosion Cracking Test

Described in ASTM G38 [111], this test is performed by prolonged exposure of C-ring (or U-bend) specimens to the applied constant load/deflection in corrosive conditions that are more extreme than the expected service ones. The absence of the crack initiation of the specimen while strained to yield means the successful qualification for the alloy product. The SCC test in Cl^−^/H_2_S + CO_2_ conditions performed as autoclave or field tests are conducted in a 25% NaCl solution equilibrated with H_2_S and CO_2_. SCC testing in Cl^−^/H_2_S + S^0^ conditions involves autoclave testing at a high pressure of H_2_S (2000–7000 psi) in the presence of free sulfur according to expected sulfur content in the field [110]. Alternatives are slow-strain rate and cyclic slow-strain rate tests that are based on the maintenance of static and superimposed static-ripple loads for plastic deformation of the specimen that is required to cause stress corrosion cracking [100]. Pre-cracked specimens are used in double cantilever beam tests where the specimen is wedge loaded and exposed to corrosive media for 720 h, followed by cleaning and measurements of the load–displacement curve with the tensile machine.

##### Hydrogen Embrittlement

This test is performed with CRAs after prolonged artificial downhole aging at elevated temperatures depending on the downhole temperatures’ requirements [110]. The test determines if the mechanically loaded to yield point, aged C-ring specimen would fail either during cathodic charging at room temperature in NACE solution due to galvanic coupling to carbon steel or upon cooling and exposure to a hot Cl^−^/H_2_S environment while being coupled to a surface of the carbon steel. The detailed technique of the test is covered in NACE MR0175 [107]. 

##### Localized Corrosion Resistance

The pitting resistance equivalent number (PREN) [100] commonly evaluates the localized corrosion resistance of the CRAs. This parameter is estimated by content (wt.% Cr + 3.3 wt.% (Mo + 0.5 W) + 16 wt.% N). The correlation of PREN to localized corrosion perceptivity is dependent on the Cl^−^ concentration and the corrosion potential and is used to classify the alloys. The alloy is subjected to long-term immersion in corrosive media and its polarization is measured starting from open-circuit potential and cycling back to the potential at which the backward potential scan intersects the forward scan, which is a common method to determine the extent of localized corrosion by circular potentiodynamic polarization [112]. Although the test is performed in the natural conditions, the time of testing may not be enough for the development of localized corrosion. An alternative test is ASTM G192 [113], where the potential is raised to a value at which pitting or crevice corrosion starts; the current is then held constant until a pre-defined charge density is attained, then the potential is decreased in a pre-determined stepwise manner until the measured current shows a monotonic decrease, indicating passivation. This way the localized corrosion is controlled by galvanostatic polarization [100]. 

### 2.4. Corrosion Control by Inhibitors

#### 2.4.1. Basic Principles

Corrosion inhibitors (CI) are chemicals that are effective in very small amounts when added to a corrosive environment in decreasing the corrosion rate of the exposed metallic materials [12]. Inhibitors could be added either continuously or intermittently [114] to production fluids, acids, cooling waters, or other fluids to inhibit the corrosion reaction. They may reduce corrosion by forming a very thin film on the metal surface, causing a passive chemisorbed layer to form on top of the metal, by forming a oxide protective film, or by removing aggressive constituents from the environment and converting it into a complex [115].

#### 2.4.2. Types of Corrosion Inhibitors

Several chemicals including inorganic complexes, organic molecules, natural products, and rare earth elements were successfully identified as corrosion preventing agents for a wide variety of metals in different corrosive environments. Inhibitors were found to be effective and unique in action depending on the metals and environmental conditions [116]. Inhibitors can be divided into two main categories—inorganic and organic. Inorganic inhibitors are mainly used in boilers and cooling towers whereas, organic inhibitors are mainly used in oil field systems. Organic CIs are fatty acids, amines, quaternary ammonium compounds, imidazolines, oxyalkylated amines, alcohols, as well as sulfur-, phosphorous-, and oxygen-containing organic molecules bearing long hydrocarbon chains and polymers [117]. The hydrocarbon chain of organic inhibitors is oil soluble; this chain provides a barrier that keeps water away from the metal surface. Figure 10 summarizes the classification of corrosion inhibitors [115]. 

The majority of oil well corrosion-preventing agents possess surfactant-like structures with polar “heads” that chelate to the metal surface and bear long non-polar alkyl chains with C_16_–C_18_ fragments [118]. This structure helps not only to form a barrier layer, but also to emulsify and solubilize the inhibitor to keep it functional at an extended range of external conditions. It should be noted that the batch of CI passes through wide variety of well conditions upon injection. It flows through temperature gradients, and fluctuations of pressure and sour gas content. Thus, chemicals in CI formulations require thermal robustness and emulsion stability at a broad range of pHs.

Inhibitors applied for corrosion prevention of well tubing are based on fatty acids, fatty imidazolines, film-forming imidazolines, and aminoimidazoles. In the wells with aggressive gases and acidic media, pH agents are injected to control corrosion via reduction of acidity; meanwhile, sour gas scavengers such as nitrites remove the aggressive action of H_2_S [119]. Oxygen scavengers including sodium sulfite, ammonium bisulfite, and sulfur dioxide, are inactive in acidic conditions and should be injected along with pH agents. Among inhibitors effective in highly acidic media, guanidine derivatives, 1-benzyl imidazole, and 2-ethyl-4-methylimidazole were reported to mitigate corrosion of low-carbon and API X65 steels in a 1M HCl solution [120]. 

For treatment of acidic wells at elevated temperatures, synthetic fatty acid triazoles and oxadiazole derivatives were proposed to maintain protection of N80 and carbon steel up to 15 wt% HCl in boiling aqueous solutions [119]. Along with inhibitors based on amines, amides, and quaternary ammonium salts, intensifiers based on acetylenic alcohols, cinnamaldehyde derivatives, formic acid, and acid-soluble salts of Cu and Bi should be injected to improve corrosion mitigation in hot conditions. Furthermore, laboratory tests of long alkyl chains bearing derived mono- and bilayer film-forming inhibitors exhibited high efficiency at HPHT, high salinity conditions in the presence of H_2_S and CO_2_.

The history of oilfield corrosion inhibitor application that started in the 1930s lead to the development of chemical formulations for a wide spectrum of oilfield conditions including operations in ordinary environments or in harsh acidizing treatments of HPHT wells. Extended action of Cis based on continuity of the films, strength of adhesion to the metal surface, as well as the prolonged release of active chemicals is crucial for the system’s integrity. Despite the importance of oil production operations, the environmental impact of the injected chemicals should be carefully considered.

#### 2.4.3. Qualification of Corrosion Inhibitors

The film-forming inhibitors’ effects are related to the formation of a molecular layer adsorbed on the metal surface; their performance is based on the persistence, continuity, and regeneration of this protective film. The qualification of CIs is performed with inhibitor-film-covered metal surfaces in a hydrodynamic flow regime mimicking the field conditions. of the corrosion rate is determined via weight loss or electrochemical measurement of surface impedance and polarization [121]. The ASTM G-170 standard [122] covers the general laboratory equipment and test methods for evaluation of corrosion inhibitors for multiphase systems. A brief description of the most common laboratory techniques were described by Sivokon [123]. It includes the static test, wheel test, bubble test, test in U-cell, rotating cylinder electrode test, rotating cage test, impingement test, and recirculating flow loop test. The static test is based on electrochemical or weight loss determination of corrosion rate in the absence of the flow of corrosive fluids. It is useful for primary comparisons and qualitative characterization of CIs.

##### Wheel Test

This test is designed for the evaluation of the efficiency of a corrosion inhibitor by the weight loss at low rate flows of the corrosive media. Flat steel coupons are placed into a wheel-shaped holder rotating inside the vessel (Figure 11) containing brine solution, CI, oil and is saturated with a partial pressure of CO_2_/H_2_S according to service conditions. Standard testing lasts for 24 h with rotation and periodical wetting of coupons in the heated corrosive fluid [123].

##### Bubble Test

This test determines the corrosion rate of the sample(s) exposed to a flow of corrosive fluid upon CO_2_ bubbling using the linear polarization resistance method. Gravimetrical measurements can also be used for the evaluation of the sample degradation [112]. The test is performed in a glass vessel with access ports for the working, counter, and reference electrodes, thermometer, and gas inlet and outlet tubes (Figure 12).

##### Test in U-Cell

In the test in U-cell generates an intense flow of hot corrosive medium saturated with CO_2_ against the metal sample in a double chamber vessel; the weight loss of the probe is compared with a control sample [123]. 

##### Rotating Cylinder Electrode (RCE) Test

The rotating cylinder electrode (RCE) test is one of the most popular laboratory methods of CI evaluation. It determines the corrosion rate of the metal sample when exposed to aggressive fluids with temperatures ≥ 80 °C at a turbulent flow by the weight loss and/or via polarization curves registration [125]. The test apparatus consists of a rotating unit driven by a motor with a controller for the rotation rate that is attached to a sample holder [126].

##### Rotating Disc Electrode (RDE) Test

The rotating disc electrode (RDE) test has a lot of similarities to RCE and varies only with shape of the electrode. Electrochemical measurements in a three-electrode cell using a rotating disc and rotating cylinder electrodes simulating laminar and turbulent regimes, respectively. Registration of polarization curves by a potentiostat with and without the CI allows for the determination of the film stability for the CI with no details regarding the influence of flow regimes. Electrochemical impedance measurements describe the efficiency of the CI both at laminar and turbulent flow [121].

##### Rotating Cage Test

The rotating cage test is performed with flat metal coupons that are installed into the sample holder at the axis of the rotating motor and dipped into the sealed cell with the corrosive fluid and gas (Figure 13) [123]. Depending on the speed of the motor rotation and dimensions of the vortex formed, various flow regimes could be implemented such as a homogeneous stream or turbulent flow [126]. The corrosion rate is determined by the sample’s weight loss.

##### Jet Impingement Test

The jet impingement test is used for corrosion surveys of the coupon exposed to the sprayed fluid. The test simulates the high-turbulence conditions at high temperatures and pressures in gas, liquid, and multiphase turbulent systems [126]. The jet impingement apparatus includes an autoclave for the fluid preparation, impingement test cell with impingement probe, a high-pressure fluid pump, and a flow meter to measure flow rates. The test cell is equipped with spray nozzles. The fluid is pumped from the autoclave through the nozzles to the probe and returns (Figure 14). The probe contains the sensor element to measure corrosion rate. The corrosion rate is determined by the weight loss and/or electrochemical measurements such as linear polarization and electrochemical impedance spectroscopy [127]. 

Important work was done by Papavinasam [128] where he compared the abovementioned methods of CI evaluation for the reproducibility between laboratory and field test results for oil and gas pipelines. He ranked the reliability of the methods from high to low as follows: wheel test, bubble test, static test, rotating disc electrode (RDE), rotating cylinder electrode (RCE), jet impingement (JI), and rotating cage (RC).

## 3. New Trends and Aspects to Consider for Development of Novel Anticorrosive Materials

Oil and gas reserves which are left to be explored and mined remain at challenging locations with extreme environmental conditions such as ultra-deep wells, arctic and highly sour reservoirs, or ‘unconventional oil’ such as shale, bitumen, and tar sands [5,118]. Such conditions imply exposure of the oil production equipment to high pressures and high- or ultra-low temperatures, and high concentrations of H_2_S, CO_2_, and chloride ions that cause extremal corrosion effects for metallic materials. Development of the novel technologies for oil production such as enhanced oil recovery (EOR) and the carbon capture and storage method (CCS) are promoting a transition to more aggressive production fluids. Thus, OCTG, especially tubulars which are in production, are subjected to ever-increasing corrosion challenges. In addition, challenging areas that have extreme environments are difficult to access so the exploration of these fields are associated with additional challenges. A summary of such factors is listed below.

### 3.1. Extreme Temperatures and High-Pressure Operations 

Exploring horizons beyond current resources, deep drilling, and development of the fields with unconsolidated sands lead to high temperatures and high-pressure exposure of downhole tubulars and equipment. The general classification of the HTHP operational conditions is given in Figure 15 [5]. Exposure of tools to extreme environments requires utilization of materials with higher mechanical properties as well as implementation of more reliable engineering solutions. Examples of such tools includes pressure-exposed blowup preventers, drilling risers, and rotary tables. In addition to the environmental challenges, advances in oil production such as Steam Assisted Gravity Drainage technology, which is used for extraction of heavy oil and bitumen from deep reserves, increases the ordinary oil production temperatures up to 150 °C [129]. In situ combustion (ISC) oil production (or fire flooding) generates temperatures up to 450–600 °C at the combustion front [129]. Moreover, elongation of the overall length of tubular structures in the deep and ultra-deep wells in addition to ocean waves at offshore platforms increases the tensile stress on the downhole tubulars. Deep water operations require new materials for production equipment that can withstand significant temperature differences between hot produced fluids and cold sea water [130]. Arctic and permafrost operations provide additional challenges of freeze/thaw damage of materials, and difficulties in joining, thermal shock, and embrittlement. Thus, for operating conditions where the temperature is below −27 °C, higher grade steels are required to prevent embrittlement and cracking. For these conditions, lightweight, fatigue-resistant, high strength materials are in high demand.

### 3.2. Aggressive Environment: H_2_S, CO_2_, and Acids

H_2_S and CO_2_ gases in combination with water are the main causes of corrosion in the oil and gas industry. In addition to this, high chloride ion content can cause very high corrosion rates and decrease production equipment durability.

Sour corrosion is caused by H_2_S exposure and results in weight loss corrosion in sour service, and pitting and sulfide stress cracking (SSC). SSC causes the highest risk for OCTG—the exposure to H_2_S at a certain level of pressure, temperature, pH, and tensile stress could result in catastrophic steel failure with very fast crack propagation. Recommendations for the steel choice depending on the severity level of the sour service is provided in Figure 16 [12,98]. Sweet and sour corrosions could occur while the steel is exposed to a mixture of H_2_S/CO_2_ and altering the ratio of H_2_S to CO_2_ in the mixture would shift the corrosion mechanism either to sweet or sour. Steel surface microstructures have a noticeable effect not only on SSC and surface roughness, but they also increase the corrosion rate in a hydrochloric acid medium for both laminar and turbulent flow regimes, so that high mechanical properties and smooth surface pipes are required for better durability [7].

Sweet corrosion caused by exposure to CO_2_ was first recognized in the 1940s in Texas [5]. Dry CO_2_ gas is non-corrosive; however, if dissolved in water, it forms carbonic acid that attacks steel, initiating pitting, mesa attack, and flow-induced localized corrosion. The average pH of CO_2_-saturated water is about 4 or slightly less. The presence of other inorganic or organic acids like HCl or acetic acid decreases the pH level even lower. Sweet corrosion causes up to 60% of oilfield failures [12].

Additionally, formation stimulation methods introduce strong acids as HCl, HF, and other mineral and organic acids to production fluids that causes equipment damage. In practice, stimulation of the oil wells is performed by pumping strong acids solutions through steel-made pumps, casing, and production tubing until it reaches the target formation to react with [131]. However, strong acids react not only with rocks but with surfaces of steel and cause its corrosion. That is why corrosion inhibitors are an indispensable part of the acidizing fluid mix; however, they cannot fully mitigate the reaction of acids with steel. Application of high permeability production fluids in reservoir stimulation are intended to help the production equipment resist these conditions and require development and implementation of reliable advanced materials.

### 3.3. New Trends in Corrosion Control Technologies

#### 3.3.1. Modern Trends in Engineering of Coatings

Novel high-performance corrosion control systems require high durability for OCTG and downhole equipment. Among the corrosion mitigation measures, organic protective coatings are the most widely used, and their costs add up to two-thirds of all anti-corrosion expenditures [132]. To achieve high performance in corrosion control, design and engineering of state-of-the-art self-healing and smart stimuli-responsive coatings are ongoing to target the action of the materials in response to the local environmental threats. The healing effects of coatings commonly functioning by restoring the integrity of coating layers by sealing defects or by inhibiting corrosion reactions at coating defects with minimal or no external physical intervention [132]. Autonomous and non-autonomous self-healing mechanisms have been developed for smart coating systems. An autonomous effect is enabled by embedding polymerizable healing agents or corrosion inhibitors into the coating matrices [132]. Non-autonomous self-healing effects are induced by external stimuli, which trigger chemical reactions. Initially, autonomous self-healing effects were developed for chromate conversion coatings and were based on the reduction of chromium(VI) corrosion inhibitors in the coating formulation to form a protective oxide film at the site of damage. Other useful inorganic corrosion inhibitors that can be incorporated into self-healing coatings are phosphates, nitrites, molybdates, tungstates, vanadates, borates, and rare earth metal salts. Not only metal-based but also organic corrosion inhibitors can be entrapped into the coating matrix including benzotriazole (BTA), mercaptobenzothiazole (MBT), imidazoline, 8-hydroxyquinoline (8-HQ), and aliphatic amines [132]. Alternatively, methods for achieving autonomous coating recovery by embedding extrinsic microencapsulated polymerizable healing agents in the coating have been engineered. Capsules rupture in response to the stimuli and release the healing agents that recover the coating integrity. Isocyanates, epoxies, and amines (such as tetraethylenepentamine (TEPA)) as curing agents are the major components of such microcapsules. 

Non-autonomous stimuli-responsive healing effects in coatings may also be facilitated by a complementary corrosion sensing component, which would detect fine pH variation or electrochemical reactions at an early stage of corrosion. Thus, conductive polymers, i.e., polypyrrole (PPy) or polyalinine (PANI), with oxidizing properties cause steel passivation. In the case of polypyrrole which has ion-exchange properties when it is modified with PMo_12_O_40_^3−^ and HPO_4_^2−^ ions, a self-healing ability is additionally achieved owing to the reaction of tetraoxomolybdate ions (MoO_4_^2−^) with iron. The iron–molybdate significantly limits substrate digestion [133]. PANI–MoO_4_^2−^ is an oxidizer which offers anodic galvanic protection to the substrate [134]. A self-healing effect based on commercial super-absorbing polymers that are used as a component of protective coatings on cold-rolled steel, can considerably reduce oxygen diffusion from solution to the surface of the protected material. For instance, AQALIC CS-7S [135] is a polymer, which can absorb water and swell. Its addition at a quantity of 5% to the polymeric coating helps to heal scratches on the coating upon contact with water (Figure 17) [136].

The recent decades have provided examples of self-healing in rigid structures such as metallic–polymeric coatings. For instance, an electrochemically produced zinc coating with polyethylene oxide-*b*-polystyrene (PEO_113_-*b*-PS_218_) nanoaggregates owes its self-healing properties to the amphiphilic properties of the polymer, which reversibly shrink and swell in a medium containing chloride ions [137]. Moreover, ceramic materials (TiC/Al_2_O_3_ and Ti_2_AlC) demonstrate efficient oxidation-induced self-healing properties. In this case, healing is activated by an increase in temperature in the presence of oxygen from the air, resulting in the oxidation of the coating components and filling of defects by formed titanium and aluminum oxides [138,139]. 

Implementation of the abovementioned stimuli-responsive and autonomous self-healing properties for industrial coating systems will drastically enrich anticorrosive performance and durability of these protective systems along with engineering of novel high-performance polymers and resins for non-metallic coatings. 

Environmental sustainability and ecological regulations have brought up importance of the utilization of natural macromolecules and biopolymers like chitosan, lignin, cellulose, nanocellulose, etc., into coating engineering. It was recently reported [140,141] that chitosan exhibits excellent film-forming, corrosion-protecting, and antimicrobial properties with very good affinity to metals. In combination with an epoxy matrix, this polymer provides synergistic effects and improved the coatings’ anticorrosive and antibacterial properties in a sustainable manner. Lignin is known as a natural UV absorber [142] that, when mixed with a resin matrix, improves the coatings’ UV stability and service life. Moreover, natural fibers could be used as carriers for healing agents in smart coatings. Thus, cellulose microfibers loaded with various amines and self-healing agents [143] were used to blend with polymeric matrices to develop smart self-healing epoxy coating formulations. Cellulose nanofibers [144] can establish partial chemical interactions with amino groups of a polyamine-based curing agent and physically adhere epoxy monomers. Upon mechanical impact, the released epoxy monomers are cured by the amine on the surface-modified cellulose nanofiber, promoting fracture closure.

It is worth noting that novel coating compositions often require additives of functional nanomaterials [145,146] for the improvement of their texture, temperature resistance, mechanical, adhesive, and wettability properties, as well as for better penetration of components in the formulation. Additives of carbon-based nanomaterials, such as CNTs, graphene, graphene oxide, and others, are known to improve mechanical stability of epoxy and other matrices [147,148]. Nano-silica enhances corrosion protection, fracture toughness, and tensile strength of coating matrices [149]. Combinations of polymeric matrices with various metal oxide nanoparticles, nano-clays, nano-ceramics, and other nanofillers results in nanocomposite coatings with superior properties vs. ordinary resins [150].

#### 3.3.2. Current Trends in the Development of Corrosion Inhibitors

Injection of corrosion inhibitors has become a prominent method of anticorrosion treatment in the oil industry, and the number of commercially available CI formulations is rapidly growing. Intense injection of chemicals, especially in open systems like offshore platforms and perforated wells, requires minimal ecological impact to maintain the sustainability of ecosystems. Therefore, environmentally friendly inhibitors based on biogenic amino acids such as methionine are of great interest for low-cost, non-toxic corrosion protection [151]. Evaluation of L-methionine’s corrosion protective effects on 309S stainless steel in 1M H_2_SO_4_ via ESI, polarization, and XPS, concluded that methionine significantly increased the charge transfer resistance, and reduced the corrosion current density due to suppression of both cathodic and anodic reactions. To further increase the anticorrosive efficiency of this amino acid and enhance protection of the metal surface, methionine-based polymers were synthesized and evaluated for CI activity [152] on mild-steel coupons in a 1 M HCl solution at 60 °C. The obtained polymers exhibited excellent inhibition efficiency due to increased adsorption onto the metal surface. Moreover, comparative evaluation of the corrosion protective action of 20 natural amino acids on iron surfaces in a 1 M HCl solution showed the best performance efficiency for sulfur-containing molecules—methionine, cystine, and cysteine [153].

Polymers possess important structural features such as better film-forming action with several possible attachment points and highly versatile derivatization, and thus have a great potential in surpassing the performance of small molecule-based corrosion inhibitors while keeping the concentration down to a minimum [154]. Biopolymers with good chelating properties such as derivatives of chitosan are emerging candidates for readily available and eco-friendly CI systems [155]. Additional possibilities of polysaccharide-backbone modifications with various side chains and heteroatoms-bearing moieties can allow the customization of the materials’ affinity to metal surfaces and film-forming behavior in a broad variety of conditions. 

Principles of chemicals’ compartmentalization and their controlled release has been applied for extended action of corrosion inhibitors, acidizing agents, and scale dissolvers. Hollow carbon nanospheres loaded with M16 yielded multifunctional corrosion inhibitor-infused porous graphite carbon nanospheres (GCN@M16) [156]. Controlled release of the M16 anticorrosive agent occurs in the response to mechanical impact and to external pressure while the hollow nanospheres provided good dispersion and lubricating actions for the obtained composite. As mentioned earlier, metal oxide nanoparticles, hollow silica, halloysite, clay, and other nanocontainers were utilized for infusion of the inhibitor. The majority of these composites were further implemented for the engineering of coatings. 

## 4. Conclusion and Future Directions

In conclusion, this work reviewed the latest industrial and environmental challenges that novel corrosion protection systems are facing in oil and gas downhole applications. The broad spectrum of the existing corrosion protection systems was reviewed and described in addition to their application features and current R&D achievements to tackle emerging challenges. A summary of the corrosion protection systems evaluation standards illustrated the scope of the tests that are currently implemented and that would require the extension of methods for assessing the improved innovative protective solutions. Expanding the horizon of oil and gas production requires the development of advanced materials that could endure highly corrosive environments with increased mechanical challenges and HPHT operations. Looking at the state-of-the-art, novel types of engineered corrosion protective materials we conclude that: smart stimuli-responsive systems for self-healing coatings, environmentally friendly formulations of corrosion inhibitors and resins, and composites based on nanomaterials infused with corrosion inhibitors and/or healing agents would beneficially address the prolonged durability of corrosion protective systems and mitigate the challenges that the oil and gas industry is currently facing.

## Figures and Tables

**Figure 1 materials-16-01795-f001:**
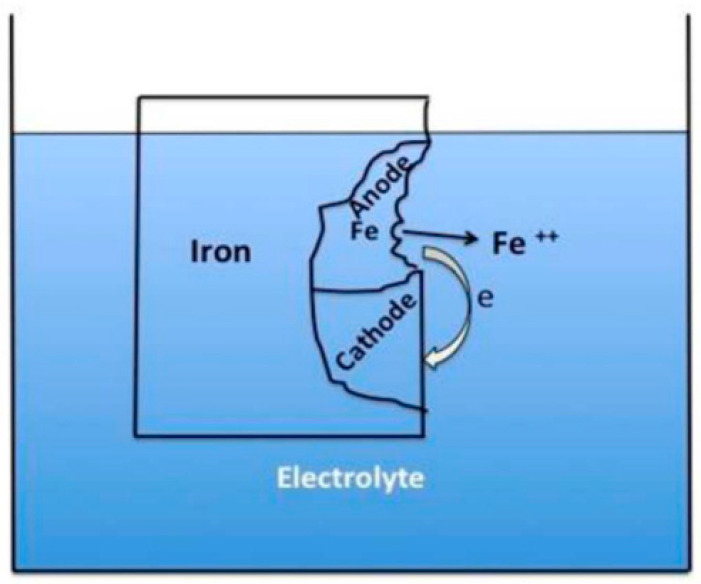
Local corrosion on steel surface [3].

**Figure 2 materials-16-01795-f002:**
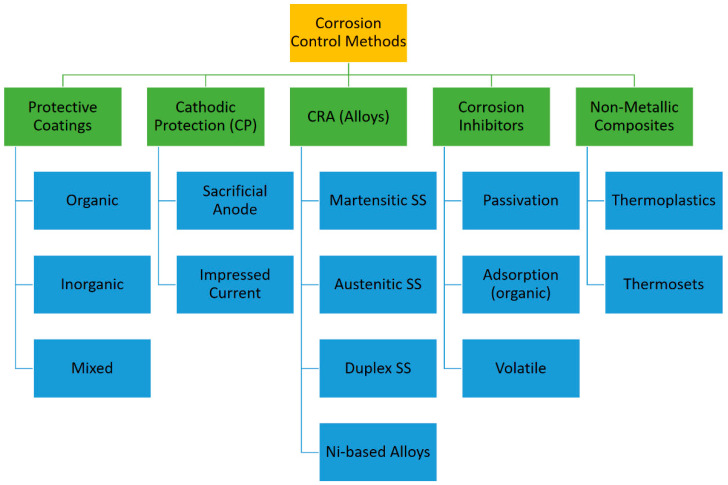
Summary on corrosion control methods.

**Figure 3 materials-16-01795-f003:**
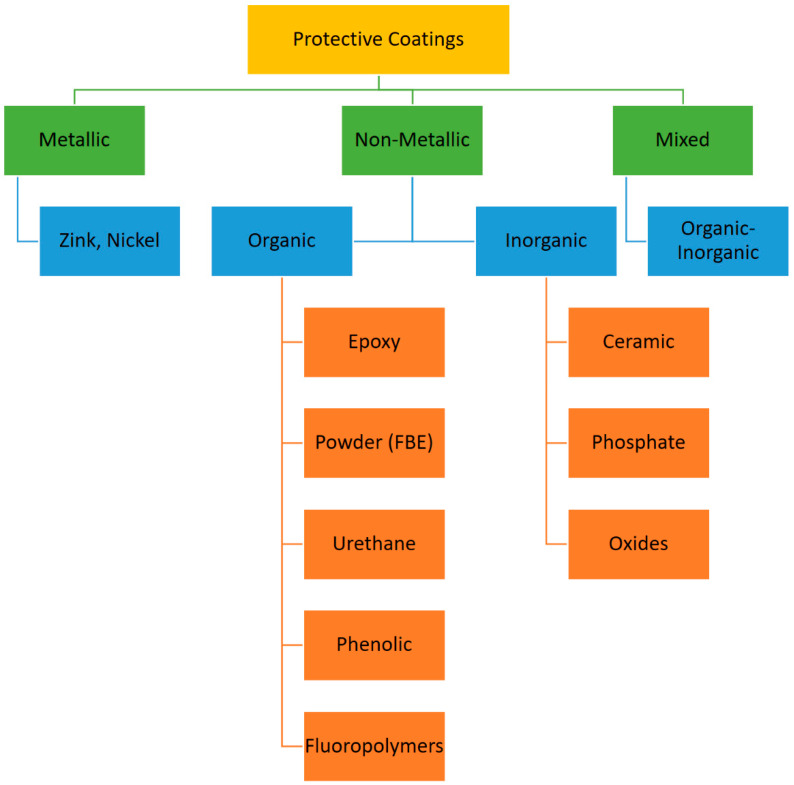
Types of protective coatings for downhole applications.

**Figure 4 materials-16-01795-f004:**
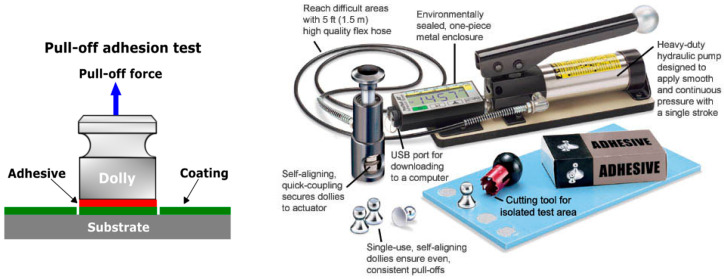
Pull-off adhesion test and required instrument kit [64,65].

**Figure 5 materials-16-01795-f005:**
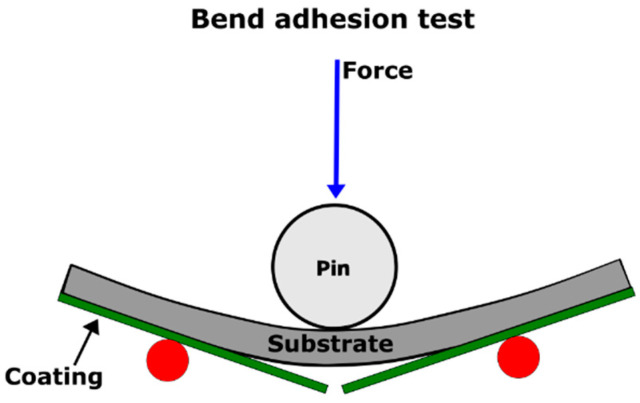
Adhesion bend test [64].

**Figure 6 materials-16-01795-f006:**
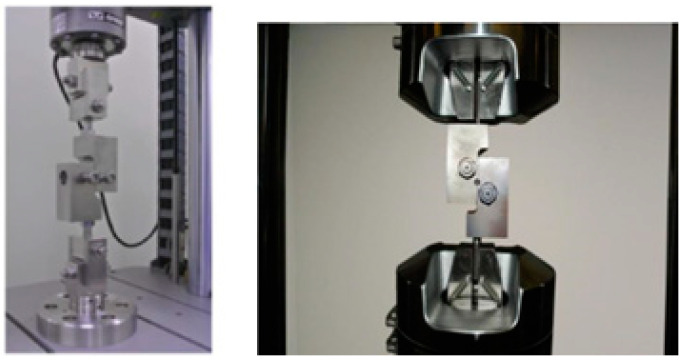
Tensile shear fatigue test [72].

**Figure 7 materials-16-01795-f007:**
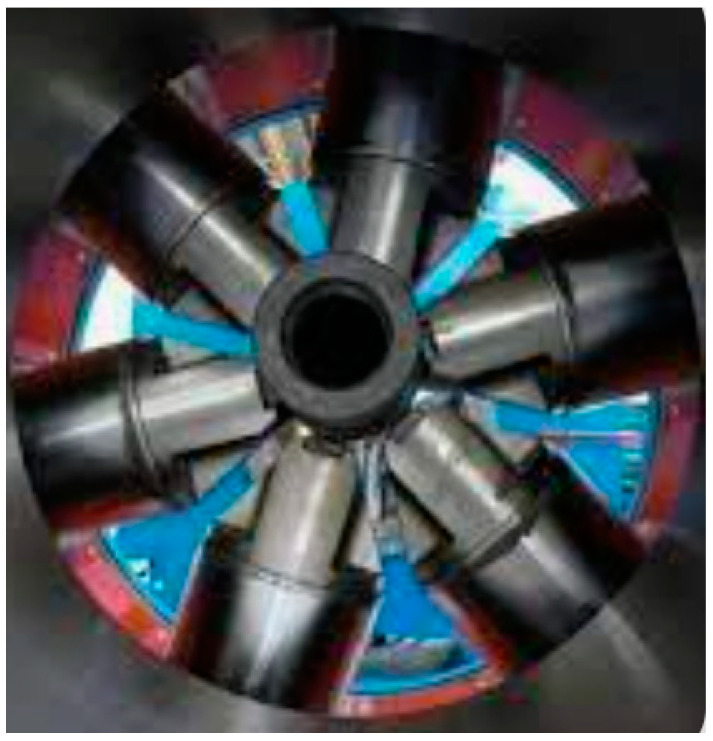
Rotational pipe torque machine [81].

**Figure 8 materials-16-01795-f008:**
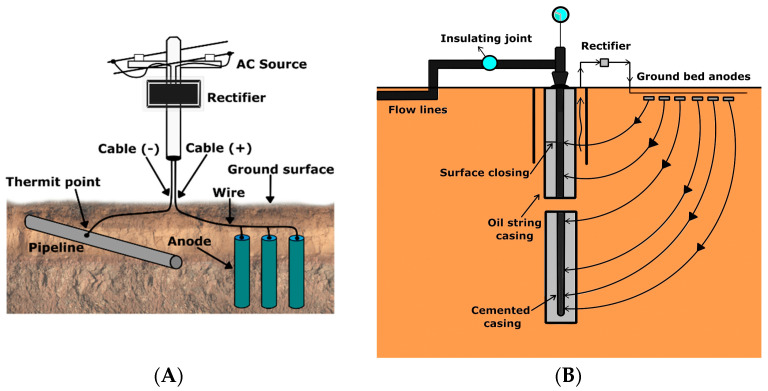
Impressed current cathodic protection: (**A**) buried pipelines [87], (**B**) well casing [20].

**Figure 9 materials-16-01795-f009:**
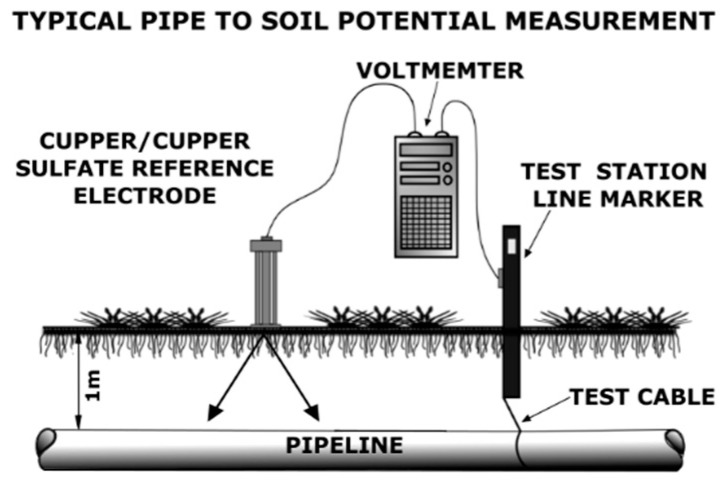
Typical procedure for structure-to-soil potential survey [97].

**Figure 10 materials-16-01795-f010:**
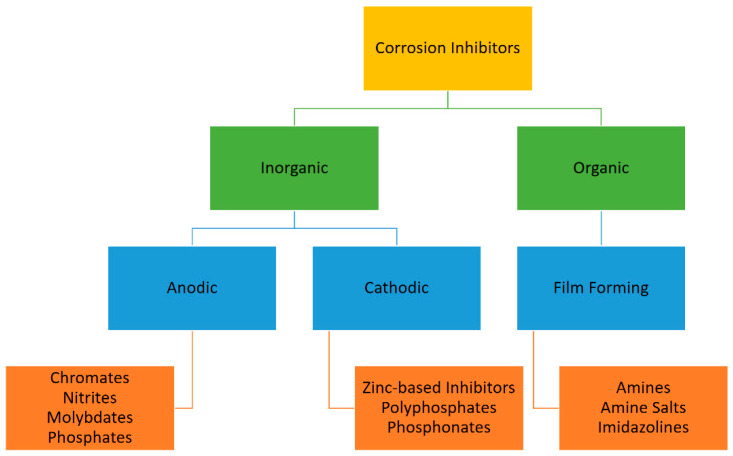
Types of corrosion inhibitors.

**Figure 11 materials-16-01795-f011:**
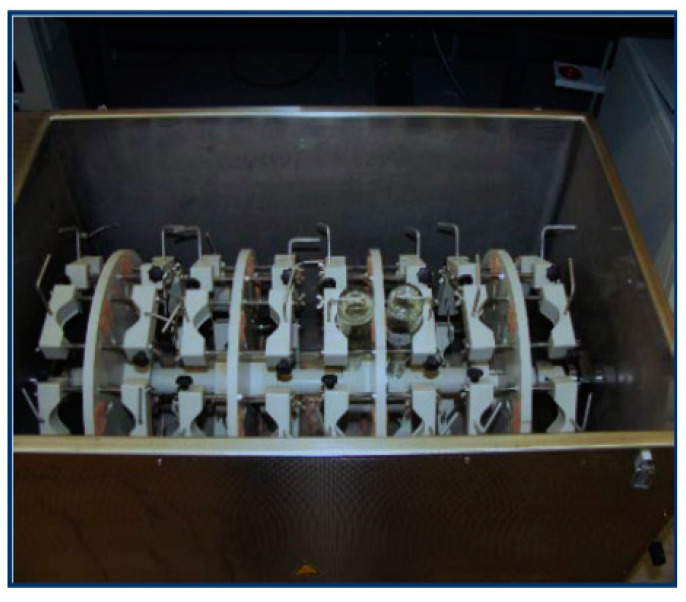
Equipment for qualification of corrosion inhibitors by the wheel test [123].

**Figure 12 materials-16-01795-f012:**
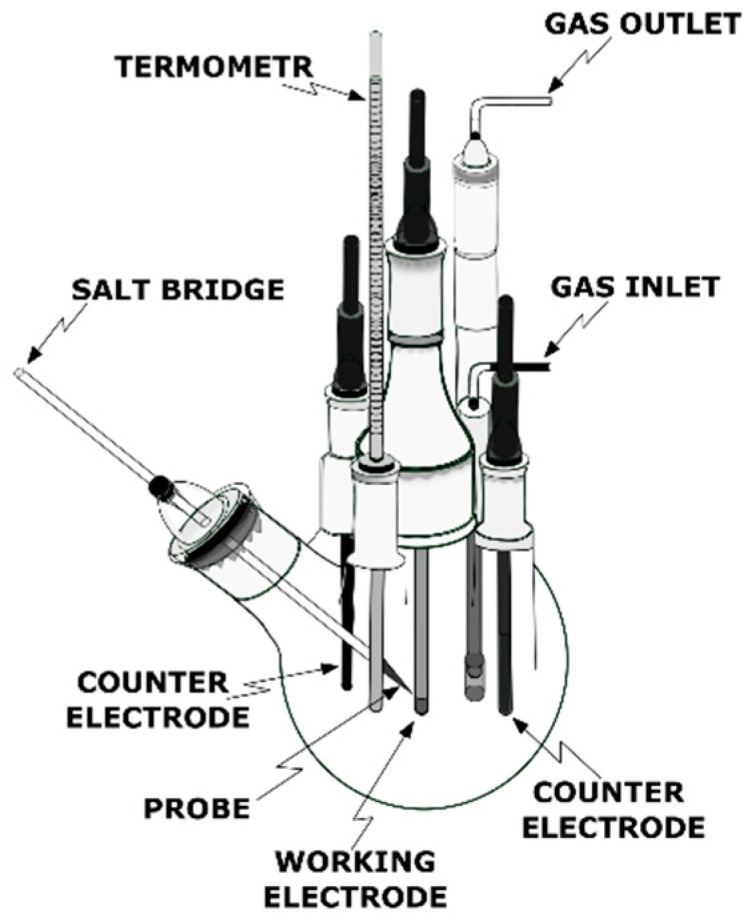
The apparatus for the bubble test [122,124].

**Figure 13 materials-16-01795-f013:**
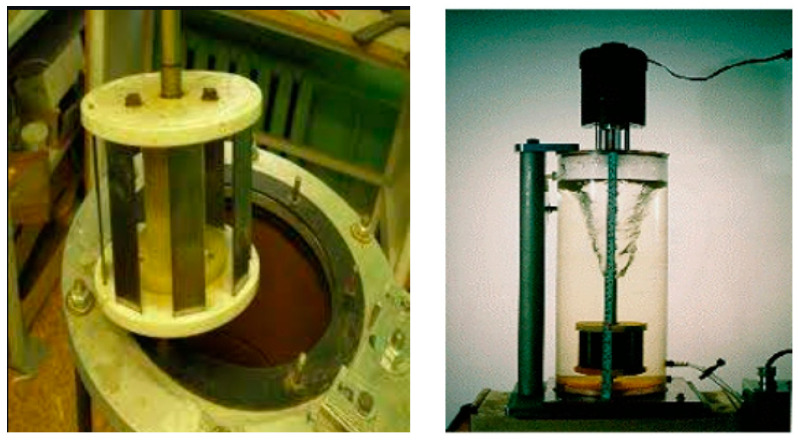
Instrument for the rotating cage test [123].

**Figure 14 materials-16-01795-f014:**
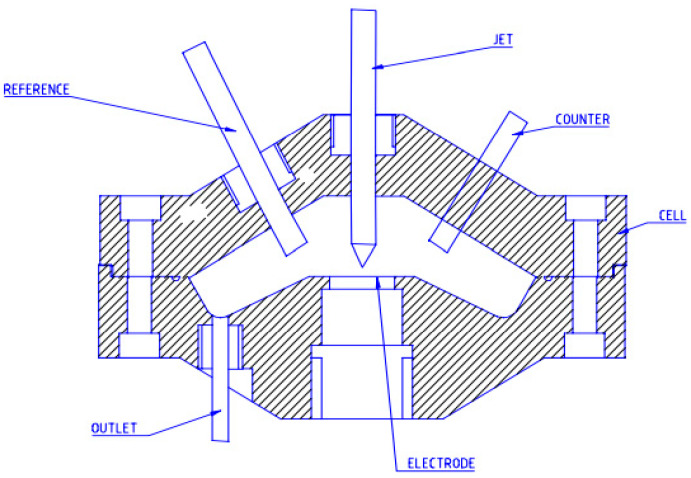
The scheme of jet impingement cell [127].

**Figure 15 materials-16-01795-f015:**
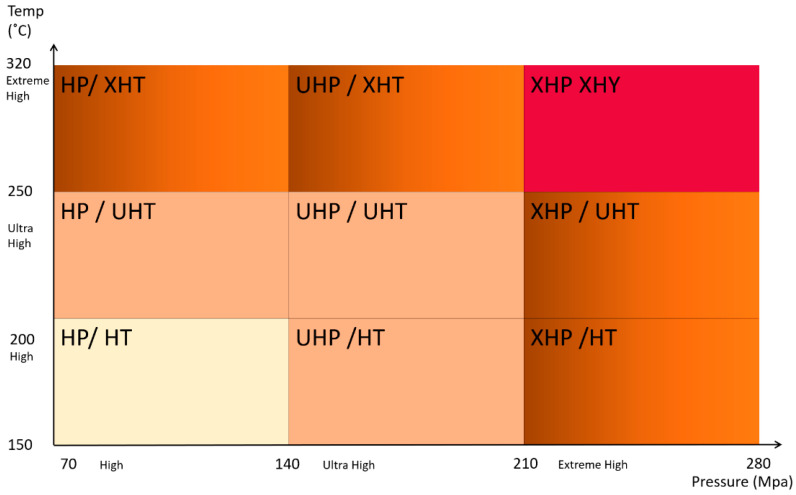
Matrix of HPHT operations representing the main operational tiers—high, ultra-high, and extreme high [5].

**Figure 16 materials-16-01795-f016:**
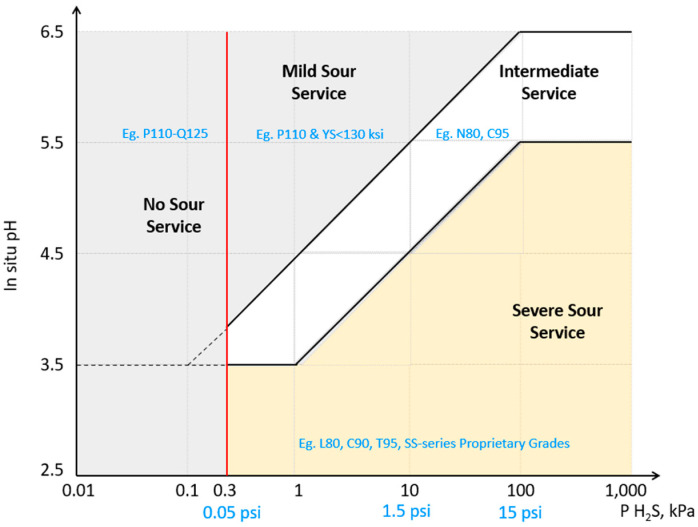
Steel materials for sour service [12].

**Figure 17 materials-16-01795-f017:**
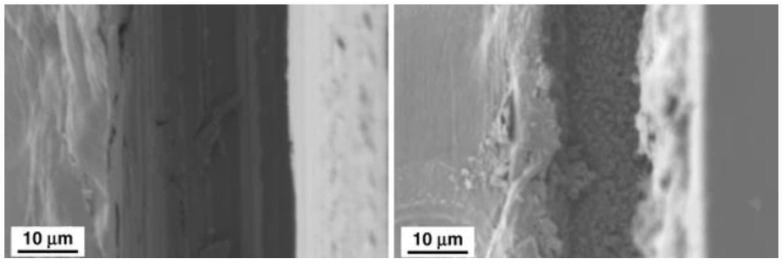
The scratched part of the coating with 5 wt% of SAP AQALIC CS-7S before and after 6 h corrosion test [133].

**Table 1 materials-16-01795-t001:** Cause of corrosion-related failure in petroleum-related industries [1].

Type of Failure	Total Failure (%)
CO_2_-related	28
H_2_S-related	18
Preferential weld	18
Localized pitting	12
Erosion corrosion	9
Galvanic	6
Crevice	3
Impingement	3
Stress corrosion	3

**Table 2 materials-16-01795-t002:** Coating systems and their application in the protection of downhole OCTG.

Coating System	Maximum Temperature, °F	Applied Thickness, µm	Primary Usage
Epoxy	175 [36]; 195 [37]	254–381 (10–15 mils) [36]	Completions [36]
Phenolic [33]	400	127–203	Production/Injection Tubing
Novolac [36]	250	254–457 (10–18 mils)	Tubing String (OCTG)
Epoxy–Novolac [33,38]	400	152–330	Production/Injection Tubing
Epoxy–Phenolic [33]	400	127–229 (5–9 mils)	Drilling/Completions
Modified Epoxy–Phenolic [33]	400	127–229	Drilling/Completions
Phenolic–Novolac [33]	350	152–330	Production/Injection Tubing
Modified Epoxy [33]	225	254–508	Production/Injection Tubing and Line Pipe
Modified Novolac [33]	300	178–381	Production/Injection Tubing
Urethanes [36]	225	127–229 (5–9 mils)	Tubing String (OCTG)
Nylon [36]	225	305–635 (12–25 mils)	Tubing String (OCTG)
Fluoropolymer [35]	350–392	No data	Tubing String (OCTG)

**Table 3 materials-16-01795-t003:** Composition of corrosion resistant steels and alloys for downhole tubing and casing [10].

Chemical Composition, Mass Fraction (%)														
Grade	Type	C		Mn		Mo		Cr		Ni	Cu	P	S	Si
		Min.	Max.	Min.	Max.	Min.	Max.	Min.	Max.	Max.	Max.	Max.	Max.	Max.
H40	–	–	–	–	–	–	–	–	–	–	–	0.03	0.030	–
J55	–	–	–	–	–	–	–	–	–	–	–	0.03	0.030	–
K55	–	–	–	–	–	–	–	–	–	–	–	0.03	0.030	–
N80	1.0	–	–	–	–	–	–	–	–	–	–	0.03	0.030	–
N80	Q	–	–	–	–	–	–	–	–	–	–	0.03	0.030	–
R95	–	–	0.45	–	1.90	–	–	–	–	–	–	0.03	0.030	0.45
M65	–	–	–	–	–	–	–	–	–	–	–	0.03	0.030	–
L80	1.0	–	0.43	–	1.90	–	–	–	–	0.25	0.35	0.03	0.030	0.45
L80	9Cr	–	0.15	0.30	0.60	0.90	1.10	8.00	10.00	0.50	0.25	0.02	0.010	1.00
L80	13Cr	0.15	0.22	0.25	1.00	–	–	12.00	14.00	0.50	0.25	0.02	0.010	1.00
C90	1.0	–	0.35	–	1.20	0.25	0.85	–	1.50	0.99	–	0.02	0.010	–
T95	1.0	–	0.35	–	1.20	0.25	0.85	0.40	1.50	0.99	–	0.02	0.010	–
C110	–	–	0.35		1.20	0.25	1.00	0.40	1.50	0.99		0.02	0.005	–
P110		–	–	–	–	–	–	–	–	–	–	0.03	0.030	–
Q125	1.0	–	0.35	–	1.35	–	0.85	–	1.50	0.99	–	0.02	0.010	–

## Data Availability

Not applicable.

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
