# Peer review of "Current Downhole Corrosion Control Solutions and Trends in the Oil and Gas Industry: A Review"

_materials, 2023, doi:10.3390/ma16051795_

Round 1

Reviewer 1 Report

The manuscript, titled: Current Downhole Corrosion Control Solutions and Trends in the Oil and Gas Industry: A Review , is a generally well-designed and useful overview of the topic presented.
However, too much emphasis is placed on some familiar basics. Of course, it is not superfluous to mention them, but it would have been more useful to address newer approaches, more examples of modern protection, and description/presentation of the currently most effective approaches in place in this area. This section is too general and not specific enough. It might be useful if the inhibition efficiency values of the newer protection methods were also given in relation to the corrosive environment (medium). In short, I would suggest that the authors expand this section. The references are mainly to book chapters, and there are few citations pf the  meaningful publications with descriptions of achievements in this field from the last five years.

Author Response

Dear reviewer,
We greatly appreciate your time and consideration of the proposed manuscript and your suggestions for the draft improvements.
We truly agree that paper have extended summary of familiar aspects and commonly utilized standards and solutions, however applied industrial practices are often based on well-known and reliable systems (that is why we have many references international standards and on books), whereas scientific research is going way ahead and promote more conservative applied research for implementation of modern achievements for higher TRL.
We thankfully addressed the notes you pointed out and we extended chapter related to the modern protections and new approaches, addressed more aspects at the modern trends such as self-healing anticorrosive systems, implementation of nanomaterials and nanocomposites as well as ecological-friendly anticorrosive solutions. Upon consideration of these modern approaches, we tried to address the correlation of effects of inhibitors for the acidity of the corrosive media and used references to highly cited peer-reviewed publications of last 5 years. We also tried to focus the extended chapter on the specific consideration of corrosion protection for the development of cost-efficient systems as coatings and corrosion inhibitors. We truly hope that our efforts to improve this draft meet expectations of revision committee.
With best regards,
Vera Solovyeva

Reviewer 2 Report

This manuscript is a comprehensive work, which mainly involves the corrosion and corrosion control in oil and gas industry.

It can be accepted after minor revision.

1.     P1, Line37 “US” can be deleted.

2.     P2, Line 49-50, The electrolyte is …… from cathode to anode. It is wrong. Electron transfers through the metal substrate.

3.     P3, Line 78-79, cathodic cracking mechanism should be Hydrogen induced cracking mechanism

4.     Fig.2, CRA, it is better to divided into Martensitic SS, Austenitic SS, Duplex SS and Ni-based alloys

Author Response

Dear reviewer,

We greatly appreciate your time and consideration of the proposed manuscript and your suggestions for the draft improvements.

We truly agree with your valuable comments and thankfully addressed the notes you pointed out

  1. P1, Line37 “US” can be deleted.

- Deleted

  1. P2, Line 49-50, The electrolyte is …… from cathode to anode. It is wrong. Electron transfers through the metal substrate.

- Thank you very much it is important note - corrected

  1. P3, Line 78-79, cathodic cracking mechanism should be Hydrogen induced cracking mechanism

- Thank you very much it is important note – corrected, mechanism is specified and descripted in more details

  1. Fig.2, CRA, it is better to divided into Martensitic SS, Austenitic SS, Duplex SS and Ni-based alloys

- Thank you very much it is important note – corrected

We truly hope that our efforts to improve this draft meet expectations of revision committee.

With best regards,

Vera Solovyeva

Reviewer 3 Report

The submitted manuscript deals with the reviewing of the Current Downhole Corrosion Control Solutions and Trends in the Oil and Gas Industry: A Review. The authors have done an extensive literature review on the subject. However, the paper needs more modifications and additions from my point of view.

1.     The introduction part is well elaborated. However, it does not accurately describe the objectives and justification of this work. Even in a review article, some issues should be described in the text. Issues: What is the need to do a review article on this topic? what is the great contribution of this article to researchers or industry?

2.     Introduction part: In the final paragraph, explain the structure of the manuscript, i.e., explain how the paper is organized.

3.     The information presented in part 2 in this paper is not very new and limited in scope. It appears to be a simple compilation of results with no critical analysis and very few comments. Thus, part 2 in this paper needs deep discussion with presenting the authors' viewpoints in this part.

4.     The third part of the paper covers New Trends and Aspects to Consider When Developing Novel Anticorrosive Materials, but the authors limit their explanation to three directions. As a result, I believe that the authors should clarify alternate directions, which would be a worthwhile addition to the paper.

5.     The Conclusion and future directions are not concise. Please write the most important conclusions in brief points. Also, please include more information about the Future Research Scope for Current Downhole Corrosion Control Solutions.

6.     The quality of figures, charts and tables is very bad. Please make effort to bring the presentation and quality of your figures, photos, and tables to the very best. The corresponding authors must contact the original authors to obtain the original data to redraw it or use their original photos to achieve the best quality in this review paper. Please make effort to bring the presentation and quality of your figures, photos, and tables to the very best. 

7.     Review papers should present novel insights or conclusions for directing the respective research area(s). Please add appropriate statements in the abstract, Introduction section, and Conclusion section.

8.     A few references need to be updated with some recent papers published in the last years.

Author Response

Dear reviewer,

We greatly appreciate your time and careful consideration of the proposed manuscript and your very detailed suggestions for the draft improvements.

We truly agree that paper have extended summary of familiar aspects and commonly utilized standards and solutions, however applied industrial practices are often based on well-known and reliable systems (that is why we have many references international standards and on books), whereas scientific research is going way ahead and promote more conservative applied research for implementation of modern achievements for higher TRL.

  1. The introduction part is well elaborated. However, it does not accurately describe the objectives and justification of this work. Even in a review article, some issues should be described in the text. Issues: What is the need to do a review article on this topic? what is the great contribution of this article to researchers or industry?
  • We address this comment in last paragraph of introduction and specified the aim of this work and our contribution on focus in the practical implementation of smart-, nano- and environmental-friendly materials for engineering of new anticorrosive industrial solutions.

  1. Introduction part: In the final paragraph, explain the structure of the manuscript, i.e., explain how the paper is organized.
  • Thank you for this suggestion – done.
  1. The information presented in part 2 in this paper is not very new and limited in scope. It appears to be a simple compilation of results with no critical analysis and very few comments. Thus, part 2 in this paper needs deep discussion with presenting the authors' viewpoints in this part
  • Thank you for this suggestion – we added several conclusions, our opinions and analytical notes/remarks for certain parts of chapter 2. We truly agree that information in this chapter is not very new, however it specifies classical corrosion protection solutions and evaluation standards extensively used in the oil industry till current days. Industrial deployment of the novel solutions is often long-lasting process and require proving examinations of novel approaches and protocols for their application. The summary of classic methods contrasts with recently engineered smart corrosion mitigation techniques developed in the laboratories and require industrial deployment.
  1. The third part of the paper covers New Trends and Aspects to Consider When Developing Novel Anticorrosive Materials, but the authors limit their explanation to three directions. As a result, I believe that the authors should clarify alternate directions, which would be a worthwhile addition to the paper
  • Thank you for this suggestion – we extended this part and draw more directions for the future development of anticorrosive materials. We additionally revised more recent peer-reviewed literature and broaden the scope of trends in the martial engineering.
  1. The Conclusion and future directions are not concise. Please write the most important conclusions in brief points. Also, please include more information about the Future Research Scope for Current Downhole Corrosion Control Solutions
  • Thank you for this suggestion – we added brief bullet-points to specify the most promising from our point of view Future Research Scope for improvement of Current Downhole Corrosion Control Solutions
  1. The quality of figures, charts and tables is very bad. Please make effort to bring the presentation and quality of your figures, photos, and tables to the very best. The corresponding authors must contact the original authors to obtain the original data to redraw it or use their original photos to achieve the best quality in this review paper. Please make effort to bring the presentation and quality of your figures, photos, and tables to the very best
  • Thank you for your note – we changed figures, tables and redraw most of pictures in our updated draft. We hope that newer version is better resolution and more contrast in colors of the text and background
  1. Review papers should present novel insights or conclusions for directing the respective research area(s). Please add appropriate statements in the abstract, Introduction section, and Conclusion section.
  • Thank you for this suggestion – we addressed this comment and specified our conclusions and insights in the abstract, Introduction section, and Conclusion parts.
  1. A few references need to be updated with some recent papers published in the last years
  • Thank you for this suggestion – we revised up to 20+ more recent literature sources for the part related to New trends with special focus on the publications of last years.

We thankfully addressed the notes you pointed out for the improvement of the first draft of the suggested manuscript. We truly hope that our efforts to improve this draft meet expectations of revision committee.

With best regards,

Vera Solovyeva

Round 2

Reviewer 3 Report

The manuscript reads well and seems better than the previous version. All my questions have been answered. I can fully recommend this work for publishing.